# Adoption Factors of FinTech: Evidence from an Emerging Economy Country-Wide Representative Sample

Khaled Mahmud [1,*], Md. Mahbubul Alam Joarder [2] and Kazi Muheymin-Us-Sakib [2,*]

1   Institute of Business Administration (IBA), University of Dhaka, Dhaka 1000, Bangladesh
2   Institute of Information Technology (IIT), University of Dhaka, Dhaka 1000, Bangladesh
*   Correspondence: khaled@iba-du.edu (K.M.); sakib@iit.du.ac.bd (K.M.-U.-S.)

**Abstract:** Adoption factors of Financial Technology (Fintech) services have been the subject of investigation in a growing body of extant literature. Macro-level as well as user-specific factors that contribute to the adoption of customer-facing fintech services have been studied. Emerging market studies mostly considered targeted demographic and socio-economic segments, limiting their ability to reflect a wide spectrum of relevant factors. We conducted a nationwide representative survey of 1282 individuals in Bangladesh. A total of 16 administrative districts from all 8 administrative divisions were included. Addressing sample imbalance with Synthetic Minority Oversampling Technique (SMOTE), we deployed Recursive Feature Elimination (RFE) to reduce number of customer features down to the most important. Using Library of Large Linear Classification (LIBLINEAR) for multivariate Logistic Regression, we identified significant features that predict customer-facing fintech adoption among individual respondents. We found that customers were less likely to adopt fintech services if they had higher reported levels of concern with security, information secrecy, limited government control, and high levels of reported service intuitiveness obstacles. Our evidence suggests these concern factors constitute the prominent factor behind fintech adoption, as opposed to demographic variables, for example. Our findings hold insights for fintech services providers and policy makers.

**Keywords:** fintech adoption; digital financial service; fintech in emerging economies; customer adoption; Bangladesh; recursive feature elimination

## 1. Introduction

Financial technologies ("Fintech" or "FinTech", hereafter referred to simply as fintech) are essentially financial services enabled by novel technological paradigms (Dorfleitner et al. 2017; Ratecka 2020). As such, the range of services that can be considered fintech is broad indeed. For example, it includes the sophisticated use of alternative credit- scoring algorithms in start-ups (Hurley and Adebayo 2016), application of Artificial Intelligence (AI) in back-office tasks in investment banking (Königstorfer and Thalmann 2020), integration of robotic financial advisory services by banks and/or startups (Baker and Dellaert 2017), among many others. For this paper, however, we narrow down the scope of fintech. Fintech, for this study, includes customer-facing financial services used by individuals which act as an alternative to and/or augmentation of traditional banking services. Fintech, in that sense, is primarily a range of banking services, enabled by technology and delivered through mobile applications. These include mobile payments, bank deposits, drawing loans, utility bill payments, offshore remittance by individuals, shopping-bill payments, and similar services availed through customer-facing platforms. Fintech platforms such as these are enabled by novel technological innovations (as well as process innovation).

A "plethora of brilliant technologies" of the fourth industrial revolution (4IR) are fundamentally transforming ways of creating value for customers (Brynjolfsson and McAfee 2016). Financial service, in fact, is one of the pioneer segments where new innovation

gets tested and subsequently commercialized for greater efficiency and profitability. For instance, the first use of computers in banking was as early as in the 1950s with Bank of America (Morisi 1996). The 4IR technologies, e.g., Artificial Intelligence, Distributed Ledger Technology (DLT), advanced computing, and mass acceptance of mobile applications enabled by high speed internet connectivity are allowing the diffusion of fintech across markets. In fact, fintech services and business models are "providing new solutions that seek to increase the efficiency, accessibility and security of financial services provision" (World Bank Group and International Monetary Fund 2019).

Fintech is poised to bring transformative changes by combining the power of digital technologies of the fourth industrial revolution (4IR) and innovative new financial services. With fintech, innovative financial solutions are delivered to users formerly either "unbanked" or "under-banked", thereby enabling communities to be financially included (Ahmad et al. 2020; Kim et al. 2018). Fintech promises financial inclusion, financial resilience, cost efficiency, better transparency, and much more (Alwi 2021; Arner et al. 2020; Beck 2020; Deng et al. 2019). Moreover, businesses can reach customers left out by traditional banks. For high-end customer segments too, fintech service providers are direct competitors of traditional banks and financial institutions. In fact, some of the more lucrative customer segments for fintech are also some of the most profitable ones for legacy financial institutions. Fintech emerged to challenge banks as the Global Financial Crisis of 2008 reduced customer trust in legacy banks (Hansen 2014; Shim et al. 2013). Success in customer value generation today depends on fintech service providers "partnering" with legacy banking firms to create a win-win situation (Ernst & Young 2019b). Banks benefit from the technological innovativeness of fintech firms, whereas startups with fintech solutions can access certain markets segment through banks without going through the full regime of regulatory and compliance hurdles.

As a result of the promised value, future of fintech holds enormous potential for all relevant stakeholders. Customers, both novel and loyal, can expect to benefit from a widening array of fintech solutions; and service providers as ecosystem enablers will benefit from expanding market opportunities. Deriving these benefits for sustainable development in some of the world's most marginalized of communities is dependent on effective adoption. Solving the technology part of the equation is just the beginning.

A large and growing body of extant literature is dedicated to the investigation of factors of effective fintech adoption, continuance intention, and customer behavior (Gomber et al. 2017; Rabbani et al. 2020; Sangwan et al. 2019; Suryono et al. 2020). The investigation of adoption factors is either from a country-level perspective or with regards to individual usage. There is evidence of heterogeneity between and within countries (Ernst & Young 2019a). Even on the macro scale, diverging strands of evidence point to differing levels of impact of variables, e.g., financial literacy (Setiawan et al. 2021). On the individual level, fintech use and adoption are influenced by a host of factors (Islam et al. 2017).There are demographic factors (Clements 2020; Imam et al. 2022; Pedrosa and Do 2011), customers' evaluation of satisfaction (Alkhazaleh and Haddad 2021; Alwi et al. 2019; Barbu et al. 2021), security risk perception, perceived ease and usefulness (Al-Okaily et al. 2021; Poerjoto et al. 2021; Wang et al. 2019). Coffie et al. (2020) finds a host of human, business, and technology-centric factors creating the environment for optimal fintech diffusion among these SMEs. Gerlach and Lutz (2017) investigated the relationship concerning demographic variables (e.g., gender, age), economic variables (e.g., disposable income) and, attitude variables (e.g., risk tolerance, knowledge regarding financial services).

Factors can also be seen from a provider-receiver perspective. Hwang and Kim (2018) divided possible factors into two dimensions. On the one side, they looked at characteristics of fintech services largely defined by service providers. These include service dimensions like complexity, underlying risk with fintech use, and trust. On the other hand, factors unique to the individual user were considered such as the user's previous experience of a security-related incident on a fintech service platform. Consistent with other studies from the extant literature, their binary logistic regression model showed negative effect of

complexity, lack of trust on service provider, and previous security experience on fintech adoption (Hwang and Kim 2018). Positive effects from users' innovative attitude were also observed. However, a systematic investigation of adoption factors for fintech in emerging economies like Bangladesh is still underdeveloped in the literature.

Bangladesh, as one of the fastest growing emerging markets, has made significant strides in terms of financial inclusion. Under the digital first policy of the government, banks and financial institutions have extended the reach of the financial system into remote areas, bringing thousands into the formal banking channel. Despite these, a sizeable portion of the population remains out of the banking system (Das 2021). Emerging fintech solutions, Mobile Financial Services (MFS) in particular, have contributed significantly to bring financial services to marginalized communities (Lee et al. 2021). However, huge work remains to enable more users to adopt fintech services. Marginalized communities in rural areas (e.g., subsistence farmers) and in urban, semi-urban centers (e.g., ready-made garment workers) can benefit from financial inclusion with accessibility and financial resilience. Adoption of fintech services, essentially through Mobile Financial Services (MFS) platforms, e.g., bKash, Nogod, Rocket, Upay would allow these communities to better access financial services for a better life.

To fight poverty, prudently manage personal finance, and access financial services for a better standard of living through fintech, it is important to understand what factors drive adoption. Fintech solutions can contribute to the financial wellbeing and resilience of users when they are open to adopting these services in the first place. Developing an understanding of fintech adoption factors of Bangladeshi customers will enable fintech service providers to better target customers and take effective marketing interventions. Such understanding will equip policy makers, financial services regulators, and ecosystem enablers (e.g., mobile network operators, investors) for effective policy directions. This study, as part of our ongoing series of research work on fintech use and sustainable economic impact for Bangladesh, takes the step towards that understanding.

In this paper, we contribute by investigating a range of demographic, economic, and qualitative factors for fintech use. Our nationwide-representative sample from Bangladesh is one of the most balanced samples used in extant literature. After incorporating a wide range of features informed by previous and existing studies, we deploy Recursive Feature Elimination (RFE) to estimate a multivariate logistic regression model for predicting fintech adoption. We find that respondents with mobile access, lower levels of reported concerns with security, and lower levels of reported geographic obstacles are more likely to use fintech services. Respondents with high levels of concerns for security and financial scam issues on fintech services, low levels of confidence using new technological solutions, and high reported levels of obstacles with service intuitiveness are less likely to use fintech services. These features add further evidence to existing literature. The contribution of this paper is mainly twofold. First, we use a nationwide-representative dataset incorporating wide demographic variation and both sides of the socio-economic spectrum. Representation of all segments were limited in previous works (Clements 2020; Coffie et al. 2021; Solarz and Swacha-Lech 2021). Second, we allow a large set of features to be selected through RFE. This enables us to input all relevant factors into the model, yet select the most important ones without possible interreference from researcher bias. Consequent findings allow fintech service providers, regulators, and future researchers to target the most salient features of individual users predicting fintech adoption.

## 2. Review of Literature

This section provides a brief summary of previous works identifying a series of macro and individual factors explaining fintech adoption intention, usage continuation, and attendant economic and social development benefits. Previous work on fintech's implication for financial inclusion is touched upon as well.

### 2.1. Security, Perceived Risk, and Trust

Perceived benefit and perceived risk from fintech use were found to significantly explain fintech adoption rates in a sample of 600 observations by Gerlach and Lutz (2021). Perceived benefit was affected by performance expectancy, economic benefits, and hedonic motivation (Gerlach and Lutz 2021). In Malaysia, most significant effects came from perceived usefulness, perceived ease of use, competitive advantage, and economic gains from use of fintech, and perceived risk factors (Cham et al. 2018). Similar results were achieved by authors in other countries with varied social, economic, and political makeups (Dishaw and Strong 1999; Hassan et al. 2022; Jibril et al. 2020; Khatun and Tamanna 2020; Mensah and Mwakapesa 2022; Musyaffi et al. 2021; Solarz and Swacha-Lech 2021; Salman and Aziz 2015). Across jurisdictions, customer experience and trust played a vital role (Amofah and Chai 2022; Cham et al. 2018; Nathan et al. 2022; Salman and Aziz 2015; Kim et al. 2015).

Nguyen et al. (2021) found that perceived security, user satisfaction, and knowledge of services were positively associated with perceived usefulness among fintech customers which influences users' continuance usage intention. Similar relationships were discovered by other authors where perceived variables mediate customer trust in fintech and thereby determine future continuous use (Poerjoto et al. 2021; Ryu and Ko 2020; Shiau et al. 2020; Wang et al. 2019). One strand of studies deals with the factors that lead to greater customer trust. For example, perceived risk and perceived benefit have been found to affect customer trust in fintech services in Islamic fintech services, and this, in turn, determines fintech adoption and usage (Ali et al. 2021). In Germany, too, data security and trust as assessed by customers have a significant influence on customer adoption of fintech services as well as user interaction with the design interface (Stewart and Jürjens 2018).

### 2.2. Literacy and Fintech Use

Laidroo and Avarmaa (2020) found higher levels of tertiary enrollment associated with larger fintech clusters. A longer span of institutional education can reliably explain financial literacy, digital literacy, and awareness of financial alternatives. Users with tertiary education are also capable of navigating the opportunities and challenges posed by fintech firms. Moreover, higher tertiary enrollment means a ready pool of talent for fintech firms. Quality talent plays a crucial role in determining how successful technology-focused start-ups will be in developing and delivering effective solutions to their target customer segments. At the same time, a steady supply of increasingly sophisticated technical talent maintains a competitive edge of fintech startups compared to regional (and global) competitors. From both demand- and supply-side perspectives, the role of higher levels of tertiary level education can be explained.

### 2.3. Perceived Usefulness of Fintech

In Jordan, perceived usefulness and enjoyment affected the intention to adopt fintech services (Al-Okaily et al. 2021). Personal variables have a role to play in determining, or at least mediating, the relationship between factors and their effect on adoption intention for fintech. And these are not entirely demographic or socio-economic in nature. For example, people with more leisure time exhibited a greater likelihood of experimenting with new fintech solutions and thus had higher levels of adoption intention. This relationship was mediated by quality of life and the level of financial literacy (Kakinuma 2022). Xie et al. (2021) discussed the factors affecting the adoption of technology by extending the Unified Theory of Acceptance and Use of Technology (UTAUT) model. They found that perceived usefulness and perceived risk along with social factors determined the adoption intention of the user. However, perceived value was another dimension they investigated. Performance expectancy, effort expectancy, and perceived risk in combination affected perceived value of the technology. Together, this had a significant effect in determining adoption intention for the technology. A similar framework can be deployed to understand the effect of these factors on users of fintech services (Xie et al. 2021).

Perceived benefit was found to have a much more significant effect than perceived risk in a study in Bahrain (Ahmed et al. 2020). Since the demographic and economic profile of customer segments affect adoption and usage intention, factors' relative importance would also logically vary across countries. In a comparative study, Mu and Lee (2017) investigated the effect of cost and service providers' credibility in determining adoption intention in China and Korea. Significant variation was observed. Cost was a major factor for Chinese customers. For Korean customers, credibility of the provider of fintech service ranked as the more significant determiner of adoption intention (Mu and Lee 2017).

In a sample collected in Hungary on Generation X fintech users, perceived usefulness, perceived ease of use, along with norms and risks related to COVID-19 explained as much as 69% of the variation in intention to use mobile payment systems (Daragmeh et al. 2021). Perceived benefits and social factors significantly affected intention to use fintech services in a survey of 500 potential fintech users during a COVID-19 time study (Nawayseh 2020). Nawayseh (2020) also found significant mediating effect of trust on the intention to use fintech services. The risk and benefits of using fintech services constitute an important determiner of adoption intention among users. The effect of these two variables depends on the customer group being investigated.

Again, following the TAM model, perceived usefulness, ease of use, costliness, and awareness were found to significantly predict of use in Malaysia. Apart from costliness, all other variables there positively affected adoption (Jin et al. 2019). Perceived benefit positively affects intention to use fintech services. Whereas perceived risk affects it negatively. However, when measured in terms of risk, i.e., financial risk, legal risk, security risk, and operational risk, effects were stronger for early adopters. Similar results in terms of risk were found by Gerlach and Lutz (2021). For late adopters, other variables were at play. Benefits and risks accounted for a small portion of the variance (Ryu 2018).

### 2.4. Demographic Factors

There seems to be a negative relationship between age and fintech use. Financial literacy helps customers access new fintech services (Hasan et al. 2022). Results from Hasan et al. (2022) confirm similar observations from multiple studies in Asia-Pacific (APAC). This underlines importance of targeted campaigns to promote greater financial inclusions for people from higher age categories. Fintech's role in promoting financial inclusion has been found significant even in the developed market. One example is from British Columbia where fintech has promoted access to new financial services for communities of underbanked people (Clements 2020). Benefits of fintech can counterbalance limitations of micro-finance facilities. As evidence from Nigeria suggests, micro-finance have their own set of limitations and systematic biases (Pedrosa and Do 2011).

In multiple studies, a resounding theme was differences in adoption rates and adoption intention across gender and age groups. Overall, fintech adoption rates were higher among the young and among males. This trend held for fintech adoption intention too, in multiple samples across markets. In SAARC[1] and ASEAN[2] markets for instance, males were ahead of females, and younger users were more likely to adopt fintech services compared to their older counterparts (Imam et al. 2022). This calls for attention of fintech service providers and regulators. To equitably distribute expected benefits of fintech, platforms need to be designed for universal appeal. Special measures would be needed to ensure women and elderly have equal access to fintech services (Imam et al. 2022). Moreover, capturing the complex interaction amid these factors calls for better measurements. In cross-country comparison, complexity of interaction among variables determining differing levels of adoption could be accounted for by better indexes (Huong et al. 2021).

### 2.5. Satisfaction and Usage of Fintech

Using Theory of Dissonance, Assimilation, and Contrast, Alwi et al. (2019) identified factors that affect customer satisfaction in fintech in Malaysia. Based on online survey results of the user of fintech services they concluded security and privacy had a very

strong influence (Alwi et al. 2019). Other factors were: information presentation, quality of service, and ease of use. Similar investigations were undertaken by others. Barbu et al. (2021) conducted similar work fintech satisfaction: testing hypothesis under Partial Least Square and Structured Equation Modeling (SEM). Fintech satisfaction is relevant not only because it determines future levels of adoption and intention to use. Satisfaction also has a spill-over effect within the larger fintech ecosystem—including for banks affiliated with fintech services platforms. In the Jordanian banking sector, for example, fintech satisfaction increases overall satisfaction for the sector (Alkhazaleh and Haddad 2021). In the last few years, fintech firms have entered emerging markets of Asia, ASEAN in particular. New players are competing with older ones. Customer segments for a large number of these firms are the same. One effective way to retain satisfaction is in a separate niche based on superior value or specialized fintech services.

### 2.6. Country-Level Evidence and Heterogeneity

Important differences across countries remain. Role of macro-level aggregates cannot be denied in determining user intention and levels of adoption at the national level. In determining adoption intention, significant country-level heterogeneity was found, both between and within countries (Kumar et al. 2021). In this case, country-level data from 30 different countries were analyzed. Adoption rates too differ significantly across countries (Ernst & Young 2019a). An intriguing observation emerged from a study in Indonesia. The model showed financial literacy to be the least significant variable in determining customer adoption (Setiawan et al. 2021). The authors also found that user innovativeness had a major role to play. They suggested greater efforts from fintech service providers and enabling regulations from the government.

Due to lower debt levels and a rising middle class, Asia as a continent was largely able to avoid the catastrophic effects of the Global Financial Crisis of 2008 and 2009. Asia's stable macroeconomic conditions have allowed the middle class to rise with increasing purchasing power. This segment now had the intention to access newer modes of services and products. These factors led to a stronger banking network inside Asia with widening coverage. Despite a growing network of banks and financial institutions, a huge population in India and China, for instance, remained outside of the banking network. Fintech has thus found the perfect ground to extend financial services to a sizeable underbanked and unbanked population in two of Asia's largest economic players. Fintech firms here can also find lucrative customer segments for every type of financial service imaginable (Alexander et al. 2017). At the same time, the role of supporting industries and ICT clusters was also identified in promoting fintech clusters in certain locations. Contrary to general expectations, however, the role of financial services clusters was subdued and not as prominent (Laidroo and Avarmaa 2020).

At the country level, fintech has important benefits for female populations. The International Monetary Fund (IMF) used cross-country data from 114 different countries and analyzed the effect of fintech in ensuring financial access for women. After accounting for endogeneity through fixed effects model, findings showed fintech to have significant economic benefits for women (Loko and Yang 2022). Benefits of greater fintech access were evident in the number of female workers in firms in countries with higher levels of fintech.

Fintech holds enormous potential for the underbanked and unbanked populations across the world (Salampasis and Mention 2018). While regulatory oversight, institutional quality, and overall macroeconomic and technological factors dictate the nature of impact, it was evident that fintech brings greater financial access and more opportunities for financial prosperity. Specific intervention strategies and commercial approaches are determined by careful consideration of these variables. Laos, Vietnam, and Cambodia have been found to hold the highest potential for fintech in the ASEAN region. These countries provide similar geopolitical, technological, political, and socio-economic makeups for fintech firms to consider (Loo 2019).

International development literature has underlined the importance of utilizing the powers of digital technologies e.g., blockchain, mobile networks, and cloud computing in changing the lives of people excluded from the formal banking channel for a while. Fintech has brought ways to realize this for some of the poorest countries in the world. Scholars are still trying to investigate the factors that lead some firms to success in extending financial inclusion for these people. One study found that among the factors, are network effects, customer centricity, the appropriateness of the commercial strategy used by the firm (Soriano 2017).

### 2.7. Fintech and Financial Inclusion

Extant literature is growing steadily in investigating fintech's implication on financial inclusion. Fintech allows for transaction disintermediation, information asymmetry reduction, new business model viability, favorable unit economics, more accessible products and services, "unbundled" service experience, the breaking of geographical and socio-economic barriers previously insurmountable, and much more (Beck 2020; Gabor and Brooks 2017; Siddik 2014). Technology-enabled platforms and business models that democratizes financial services, traditionally unavailable for low-income individuals/households, should have a positive impact. However, a systematic investigation of exactly how fintech achieves this, and indeed whether this expectation holds ground for users of all backgrounds and across markets is in order. Philippon (2019) illustrated the point with two cases: robo-advisory and alternative credit rating. With the former, it was clear that fintech enables lower-income households to access wealth management services historically reserved only for wealthy households. The latter addresses existing inefficiencies in credit scoring. It reduces "non-statistical" biases inherent in traditional processes (Philippon 2019). Alternative credit scoring used by fintech platforms allow non-traditional customers to access credit at lower cost. This is specially the case for "invisible primes": people with low credit scores and almost no credit histories (Di Maggio et al. 2022). Present in both of aforementioned cases are transaction disintermediation and cost-efficiency contributions of fintech.

Use of alternative datasets allow relatively "risky" customers to be eligible for credit, customers who might otherwise have been labelled as "subprime" (Jagtiani and Lemieux 2019). Indeed, prospects of fintech in association with state and interstate development apparatus cannot be overlooked. Fintech's predictive power for example allows the incorporation of behavioral financial factors, thereby allowing services providers to better know their "irrational customers" (Gabor and Brooks 2017). However, to what extent fintech facilitates financial inclusion at the systemic level is open to question still. The ability of fintech service providers to venture into markets banks have to get out of and/or find no longer profitable, is an advantage. It is beneficial for customers who are enabled by their services. Yet entry into ever riskier segments, adjusted for the technological superiority and attendant predictive power, may have contradictory effects on specific market segments and overall financial system. In fact, an Asian Development Bank working paper raised important questions. While presenting a series of new opportunities fintech presents for customers and regulators, the paper also notes unique new challenges. Where should the trade-off be between an open, transparent, a more "unbundled" financial services ecosystem on the one hand and a regulatory framework that is invariably required to keep the ecosystem healthy for all participants on the other (Beck 2020)? This is one of the questions that needs to be addressed.

## 3. Materials and Methods

### 3.1. Dataset

A nationwide representative survey collected data between April 2022 and June 2022, covering 2 districts from each of the 8 administrative divisions in Bangladesh. Each district was selected from either side of the poverty spectrum. Eighty responses were collected from each district. This resulted in 1282 fully completed responses (See Table A1, Appendix A for sample selection method, list of districts, and stratification process). After data wrangling

and cleaning, our sample size stood at *N* = 1036. To gain a comprehensive view of fintech usage and related factors, the questionnaire covered a wide number of demographic, economic, behavioral, technological, and perceptual/opinion-related questions. Appendix B provides a full list of variables used for our modelling purpose. Steps of our study method is visualized in Figure 1a,b shows choropleth of sampled districts.

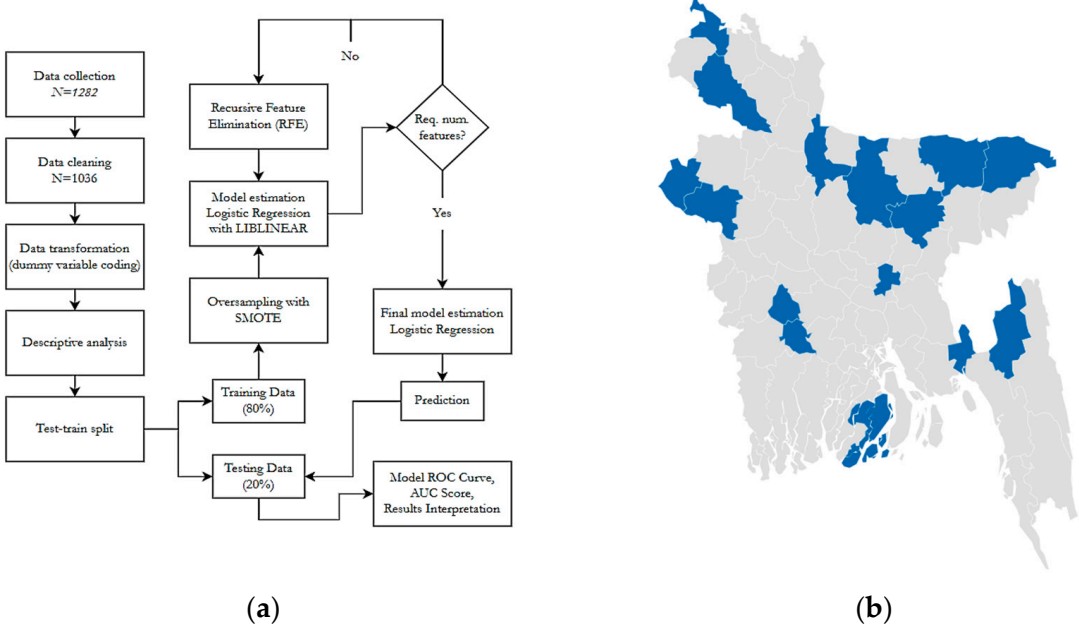

(**a**)           (**b**)

**Figure 1.** (**a**) Flow diagram of the current research; (**b**) Choropleth map of selected districts in the current sample.

### 3.2. Dataset Train-Test Splitting and Oversampling

In our original dataset (after cleaning) of 1036 instances, 29.25% were fintech users. Fintech user was defined as respondents with a minimum monthly frequency of using a fintech service of two. Before training the logistic regression model, the dataset was split into training- and test-sets with 0.8:0.2 ratio. Since the two classes of the dependent variable were imbalanced, we used Synthetic Minority Oversampling Technique (SMOTE) to increase number of instances in the minority class (non-users in this case).

In classification problems, one of the challenges researchers face is non-availability of instances across classes uniformly. The result is asymmetry across classes. Model results based on asymmetric classes may lead to inaccurate predictions with potential serious results depending on the use of the model. As an example, false negatives due to model inaccuracy that can be traced to sampling asymmetry may lead to serious health hazards (Chawla et al. 2002). To address the issue, a number of techniques have been deployed in the literature. SMOTE oversamples the minority class to bring symmetry. The method has widely been used in computer science, software development, biological classification, and more (Amirruddin et al. 2022; Ijaz et al. 2018; Pears et al. 2014).

SMOTE is widely used in multiple disciplines due to its ease of use and effectiveness in a wide range of scenarios. The algorithm works by choosing a required number of "k-nearest neighbors" for each instance in the minority class and apply linear interpolation to randomly generate instances until the class imbalance is addressed. The synthetic instances are added and a new dataset is then constructed. Since random selections of minority class data are not repeated, SMOTE successfully avoids model overfitting issues (Chawla et al. 2002). Figure 2 shows that the imbalance problem has been addressed through SMOTE and ratio of fintech users and non-users in the training set is 1:1.

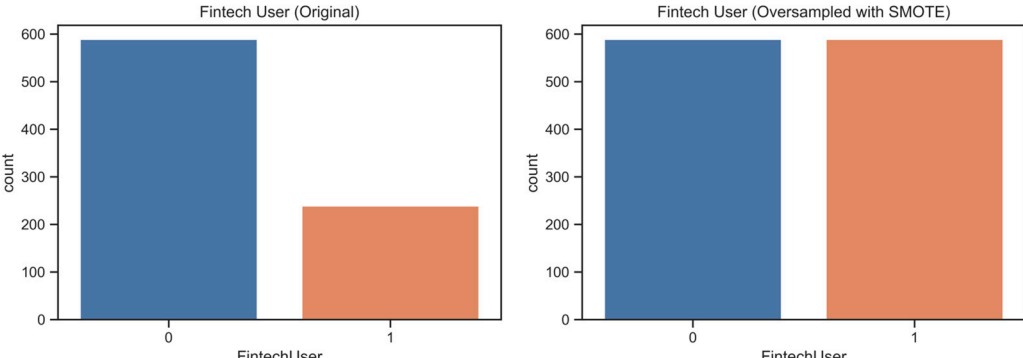

**Figure 2.** Distribution of fintech users and non-users in the original set after split (**left**) and distribution after SMOTE (**right**). Fintech users are denoted by "0" (in blue) and fintech users are denoted by "1" (in orange).

### 3.3. Recursive Feature Elimination (RFE)

In our dataset, there were 133 features. In order to reduce the number of features to a desired level, we used Recursive Feature Elimination (RFE) to obtain the most important 55 features. RFE is an iterative process of fitting and refitting a machine learning model until a desired number of features with the highest-ranking scores are retained as final estimation. RFE can be implemented for a wide range of models including Linear Regression, Logistic Regression, and Random Forests. The initial model is estimated using all features specified. In each step, the algorithm calculates a performance score, known as variable ranking, for all included features. Each successive step consists of elimination of lower ranking variable(s) and re-estimation of the model with remaining features (Kuhn and Johnson 2013). The process continues till a specified number of features is reached and the best fitting model is retained.

Success of RFE depends largely on classifier used and its relationship with the underlying loss function. Li and Yang (2005) found that ability of classier to "penalize[ . . . ] redundant features and [to] promote[ . . . ] independent features" during the iterative process contributes to its success. RFE has been used in a wide range of disciplines including bioinformatics, clinical studies, early diagnosis of cancerous cells, as well as in computer vision and natural language processing (Basak et al. 2021; Bedo et al. 2006; Bursac et al. 2008). Although computationally intensive, RFE offers one major advantage over manual feature selection. In logistic regression in particular, and regression modelling in general, the key challenge is the selection of some combination of variables/features while eliminating others (Bursac et al. 2008). Selection may prove difficult when a large number of features are involved and related theoretical supports are under development. In this case, RFE aids feature selection by prioritizing model fit.

### 3.4. Logistic Regression

Logistic regression is an econometric tool used widely for classification and predictive modeling. In binary logistic regression, the dependent variable assumes either 0 (e.g., failure, absence, negative etc.) or 1 (e.g., success, presence, positive). In this study, the dependent variable is binary, where 0 indicates no use of fintech and 1 indicates use of fintech services. Logistic regression as a method has been deployed by a number of authors for the investigation of fintech adoption. After using a binary logistic regression for intention to shift to fintech services among German households, it was found that young people had higher probability of shifting to new fintech services compared to their older counterparts (Jünger and Mietzner 2020). This relationship between age and fintech usage intention and adoption in general confirms similar findings in emerging markets as well. Jünger and Mietzner (2020) also found an interesting relationship between consumer-assessed need for transparency and probability of fintech usage. Users who had higher emphasis on transparency in banking activities were more likely to adopt fintech services.

Their logistic regression models also find households with higher levels of fintech expertise more likely to adopt fintech services (Jünger and Mietzner 2020).

Solarz and Swacha-Lech (2021) undertook a more comprehensive dataset. Their logistic regression model used a variety of demographic and attitude features with $N = 1236$. Data were collected from Poland. Findings revealed that high-income millennials with tolerance towards technological novelties were more likely to adopt fintech services. Moreover, logistic regression as a methodical approach has not been limited to individual fintech adoption behavior. Country-level investigation, such as by Okoli and Tewari (2020) for 32 African economies during 2002–2018 and by Zarrouk et al. (2021) in the United Arab Emirates (UAE), through multivariate logistic regression, also yields important findings. Role of structural support systems (i.e., regulatory support, availability of complementary resource bases) was highlighted as aid to fintech adoption.

In logistic regression, the hypothesized probability of occurrence is determined by the Sigmoid function (Dougherty 2011):

$$p_i = F(Z_i) = \frac{1}{1 + e^{-z_i}} \tag{1}$$

where as $Z$ tends to infinity, $e^{-z}$ tends to 0 and $p$ has an upper bound of 1. Conversely, as $Z$ tends to minus infinity, $e^{-z}$ tends to infinity and $p$ has a lower bound of 0. Figure 3 depicts a hypothetical sigmoid function. In multiple logistic regression, $Z$ is dependent on a vector of observed covariates $X_i$ and a linear function of these covariates with coefficients $\beta_i$:

$$Z_i = \beta_0 + \beta_i X_i + \varepsilon \tag{2}$$

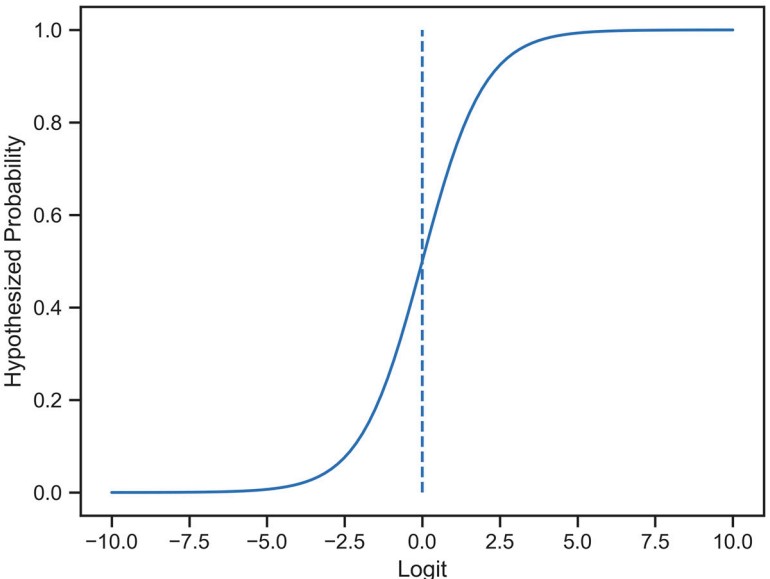

**Figure 3.** Hypothetical sigmoid function on the coordinate plane.

For our estimation purposes, the dependent variable was fintech adoption, binary coded 0 and 1 for non-adoption and adoption, respectively. We defined fintech adoption as "yes" when a respondent's frequency of using any fintech service over the last one month was greater or equal to 2. Table 1 provides a summary of dependent and independent variables in our logistic regression model. A full list of variables used is given in Table A2, Appendix B.

**Table 1.** Dependent and Independent Variables for Multivariate Logistic Regression.

| Variable Class | Variable Name | Labels |
|---|---|---|
| Dependent | Fintech adoption | 1 if frequency of monthly fintech of use during the preceding month is ≥2; 0 otherwise |
| Independent/Predictor | See Appendix B for full list | - |

### 3.5. Model Estimation with LIBLINEAR

Selection of optimization/solver algorithm for logistic regression estimation is influenced by dataset characteristics, research methods, and underlying advantages/disadvantages of the algorithms themselves. Widely-used solver algorithms include: Newton's Method, Library for Large Linear Classification (LIBLINEAR), Stochastic Average Gradient (SAG), Limited Memory Broyden–Fletcher–Goldfarb–Shanno algorithm (L-BFGS) (large scale bound constrained algorithm). For our purposes, we deployed LIBLINEAR. This was done both in feature elimination with RFE and final model estimation. LIBLINEAR is an open-source, easy-to-use package for large scale linear classification (Fan et al. 2008). LIBLINEAR has been proven to outperform other modelling algorithms not just for linear modelling scenarios. For non-linear estimation purposes, LIBLINEAR was found to be computationally less intensive (Fan et al. 2008). The result is shorter estimation time and better model fit. There have been updates on the original class of large scale linear classification algorithms; in many applications these have been found to reach accuracy equal to non-linear classification methods (Yuan et al. 2012).

### 4. Results and Discussion

#### 4.1. Description of Sample and Fintech Use

A large number of variables in the dataset were categorical. Numerical data was collected, as well. Table 2 presents descriptive statistics on numerical variables included in the logistic regression for this paper. A number of statistical analyses was performed for both numerical and categorical variables including cross-tabulations, chi-2 test of independence among groups, and correlation analysis among the numeric variables. The following sections summarizes some of the insights gathered.

**Table 2.** Descriptive Statistics of Numeric Variables.

| Variable | Obs | Mean | Std. Dev. | Min | Max |
|---|---|---|---|---|---|
| Age | 1036 | 39.758 | 13.018 | 18 | 85 |
| Household | 1036 | 5.02 | 1.768 | 1 | 12 |
| Expenses | 1036 | 15,511.486 | 8003.895 | 2000 | 70,000 |
| ExpRent | 1036 | 430.106 | 1609.094 | 0 | 16,000 |
| ExpFood | 1036 | 8740.287 | 5840.052 | 0 | 85,000 |
| ExpUtilities | 1036 | 962.074 | 992.533 | 0 | 7000 |
| ExpEducation | 1036 | 1682.082 | 2397.988 | 0 | 30,000 |
| ExpHealthcare | 1036 | 1118.972 | 1649.737 | 0 | 20,000 |
| ExpEntertainment | 1036 | 217.693 | 347.913 | 0 | 1500 |
| ExpClothing | 1036 | 958.605 | 875.719 | 0 | 7000 |
| ExpHouseHelp | 1036 | 83.966 | 521.206 | 0 | 8000 |
| ExpMisc | 1036 | 1260.523 | 1523.562 | 0 | 10,000 |
| Income | 1036 | 18,372.201 | 10,952.253 | 0 | 100,000 |
| AnnualSaving | 1036 | 11,334.555 | 32,170.068 | 0 | 450,000 |
| BankVisit | 1036 | 0.433 | 0.918 | 0 | 5 |
| Data usage | 1036 | 5553.042 | 13,071.003 | 0 | 90,000 |
| Max fee per 1000 | 1036 | 8.145 | 3.862 | 0 | 20 |

### 4.1.1. Demographic Variables

The respondents in our survey were aged between 18 and 85 years. In total, 75% of the respondents were aged below 47. Male respondents accounted for around 86% of the respondents. We observed significant variation in gender-disaggregated age across fintech usage. Taking house-type as a proxy for urban/rural area of the respondent, fintech user was found to be more prevalent in urban areas, as expected.

### 4.1.2. Economic Variables

Due to incorporating sixteen districts from the country and both sides of the poverty spectrum, our survey consisted of a wide range of income. In fact, income was significantly skewed to the right with a few extreme positive values. Fintech usage was concentrated mostly around the middle of the spectrum. As shown in Figure 4c, fintech usage was more prevalent among young users with a minimum level of monthly income.

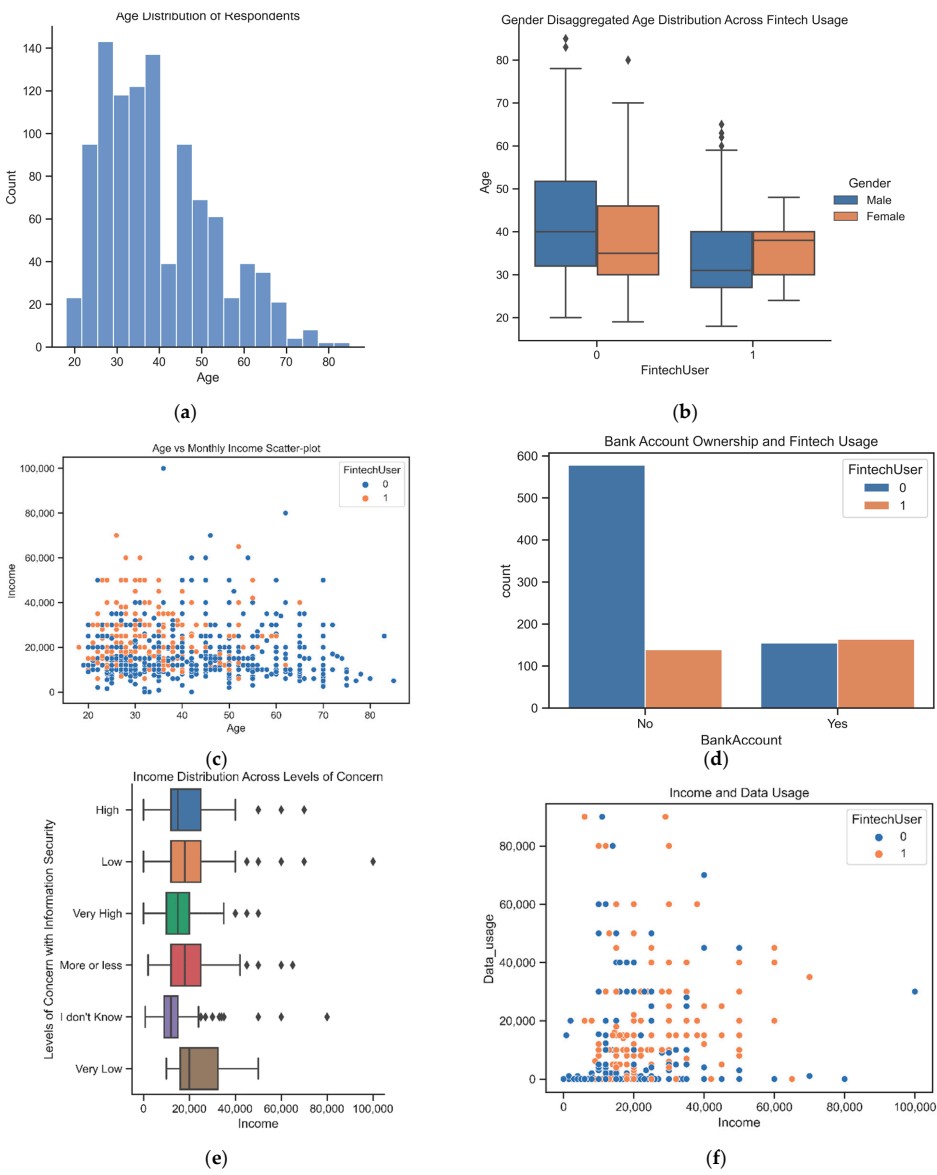

**Figure 4.** (**a**) Distribution of respondent ages; (**b**) Gender-disaggregated box-plot of age across fintech use; (**c**) Scatter-plot of age and monthly income (fintech use more prevalent in younger and high-income areas; (**d**) fintech use and bank account ownership; (**e**) Distribution of monthly income across different levels of concern for information security; (**f**) Scatter-plot of monthly income and data usage.

### 4.1.3. Bank Account Ownership

In our original dataset, 30.79% owned a bank account while the rest did not have bank accounts. However, 68.83% owned mobile banking accounts and around 31.17% did not. Lack of bank account ownership was more prevalent in rural/semi-urban areas compared to urban areas, as expected. The difference was less pronounced for mobile banking account ownership. Account ownership was also found relatively higher in higher classes of institutional education.

### 4.1.4. Internet Usage

Of the respondents surveyed in the original dataset, 36.97% had access to the internet. Of female respondents, only 21% had access to internet. The share of male respondents with access to internet was 40%. Average data usage per month (measured in megabytes) for the entire sample was 5553.042 MB.

### 4.1.5. Concerns Related to Fintech Usage

We included a wide range of variables to measures respondent's level of concern on a five-point Likert scale. These included concern for financial scandal, information security, information secrecy, limited government control, and cashless community, among others. Variations were observed in levels of concern in fintech usage across age, gender, levels of education, and occupation. In general, concerns were higher in older age groups and lower in higher income groups in our dataset.

### 4.1.6. Mental Preparedness for Fintech Usage

Respondent mental preparedness was measured on a five-point Likert scale. Respondents who used fintech services reported relatively higher levels of mental preparedness. Conversely, low reported mental preparedness was observed more frequently among non-users. Mental preparedness, on average, was lower in higher age groups, as expected from evidence in the literature. Table 3 provides a cross-tabulation of mental preparedness and fintech use. Younger users were generally ahead in technology adoption and openness to using new technological solutions. There are extreme older instances in lower levels of mental preparedness to use fintech.

**Table 3.** Crosstab of Mental Preparedness and Fintech Use.

| Mental Preparedness | Fintech User | | |
|---|---|---|---|
| | **0** | **1** | **Total** |
| Not prepared at all | 26.06 | 2.64 | 19.21 |
| Low prepared | 31.65 | 31.35 | 31.56 |
| Average preparedness | 30.29 | 44.88 | 34.56 |
| Prepared | 10.64 | 16.83 | 12.45 |
| Adequately prepared | 1.36 | 4.29 | 2.22 |
| Total | 100.00 | 100.00 | 100.00 |

### 4.1.7. Obstacles, Affordability, and Costliness

We expected higher levels of reported perceived obstacles to be associated with low probability of fintech use. Association was also expected in terms of perceived affordability and perceived costliness of the fintech service. The reason both "affordability" and "costliness" was used in the survey is due to a nuance between the two terms in Bengali, which was medium of instruction for the questionnaire. Costliness is an impersonal assessment of how expensive the service is. Affordability, on the other hand, is a more personal connotation, and respondents evaluate how easily they can access the fintech service. We observed significant variations in levels of reported obstacles, affordability, and costliness of fintech

services across demographic and behavioral categories. Table 4 provides a tabulation of affordability and fintech use.

**Table 4.** Tabulation of Affordability and Fintech Use.

| Affordability Perception | Fintech User | | |
|---|---|---|---|
| | **0** | **1** | **Total** |
| I don't know | 23.19 | 0.99 | 16.70 |
| Not affordable at all | 0.95 | 0.66 | 0.87 |
| Not affordable | 26.88 | 14.52 | 23.26 |
| Neutral | 34.11 | 65.02 | 43.15 |
| Affordable | 14.46 | 17.16 | 15.25 |
| Highly affordable | 0.41 | 1.65 | 0.77 |
| Total | 100.00 | 100.00 | 100.00 |

### 4.2. Logistic Regression Results

Model parameters, feature coefficients, and *p*-values for our final logit model is presented in Appendix C. We note that McFadden's pseudo R-squared for our model is 0.677, which is satisfactory considering the target variable is a complicated social and behavioral phenomenon. Figure 5 depicts Receiver Operating Characteristics (ROC) Curve and the AUC score of the estimated logistic regression model. For our model, AUC score was 76.22%. ROC Curve can be summarized as a plot of the sensitivity and specificity, where true positive rates are plotted against false positive rate. Area Under Curve (AUC) is a single metric summary of usefulness of the model from ROC perspective. Generally, an AUC score of 0.50 is of no use, as it indicates no better results than a random guess. AUC scores between 0.7 and 0.8 are regarded as acceptable; scores between 0.8 and 0.9 are regarded as good; and those above 0.90 are regarded as "outstanding" (Mandrekar 2010).

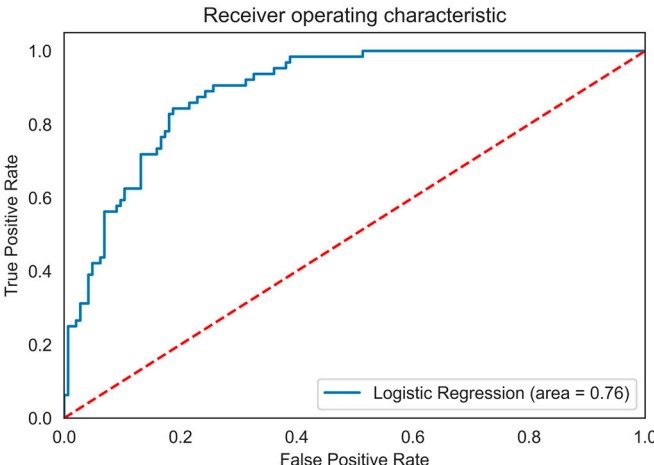

**Figure 5.** Receiver Operating Characteristic (ROC) curve and Area Under Curve (AUC) score for the estimated model.

ROC is widely used to assess the fit of a diagnostics test across disciplines. In biomedicine, tolerance for an acceptable AUC score is generally high. Relying on tests with low scores can prove to be fatal (Cook 2007; Jones and Athanasiou 2005). However, complex social phenomena are affected by a host of factors with possible interactions among them. Moreover, human behavior is involved in a target variable like fintech use. In these scenarios, an expected model AUC score above 0.9 may not be warranted. In fact, Jagtiani and Lemieux (2019) deployed machine learning to evaluate the role of alternative datasets in fintech lending platforms with logistic regression analysis with AUC scores in

the range of 58.74% to 68.88%. Higher AUC scores were achieved by Huang et al. (2020) with combinations of scorecards and the Random Forest model. Even in that case, the highest AUC score was 0.84. Table 5 below shows the accuracy scores. Weighted average precision, recall, and f1-score for our model were all 80%.

**Table 5.** Classification report table for the estimated model.

| Outcome | Precision | Recall | f1-Score | Support |
|---------|-----------|--------|----------|---------|
| 0 | 0.85 | 0.87 | 0.86 | 144 |
| 1 | 0.69 | 0.66 | 0.67 | 64 |
| Accuracy | | | 0.80 | 208 |
| Macro Avg. | 0.77 | 0.76 | 0.77 | 208 |
| Weighted Avg. | 0.80 | 0.80 | 0.80 | 208 |

*4.3. Discussion*

4.3.1. Theoretical Contribution

This paper makes a number of theoretical contributions. First, we use a nationwide representative dataset which allowed us to consider demographic, socio-economic, and geographic variations in the target user base. Because of our large sample, constructed from rigorous stratified sampling, this study was able to investigate fintech adoption from a macro perspective. Previous studies employing machine learning on primary datasets have mostly worked with smaller sample sizes (Di Maggio et al. 2022; Hwang and Kim 2018; Mukherjee and Badr 2022; Sharma et al. 2021). Second, we use RFE on the original dataset containing 133 features in total. RFE is an automated feature elimination process based on model accuracy at each iteration. This allowed us the option of not having to specifying a weighing scheme for domains of variables, thereby possibly avoiding researcher bias and/or limitations of existing theory. Third, previous work, for instance by Carlin et al. (2017), Ryu (2018), Chen et al. (2021), showed there were differences across age and gender groups regarding fintech's implications and adoption. In fact, a large number of studies in the literature investigates fintech use, usage intention, and effect of fintech across demographic groups. Contrary to these findings, our model shows little evidence of any significant effect of demographic variables when it comes to fintech adoption. Instead, the main factors that determine fintech adoption are related to customer perception of risk, costliness, and obstacles, among other things.

Through Recursive Feature Elimination (RFE), we obtained 55 important features from the 133 fed into our model. These were used to estimate a logistic regression model with fintech use as the dependent variable. As shown in Table A3 in Appendix C, of the 55 features selected, a total of 26 were found significant at the 5% level. Interestingly, most of these are related to customer-reported levels of concern, obstacles faced, satisfaction, and costliness with fintech use. We also observe that almost none of the demographic and economic variables fed into the model turned out to be significant predictors of fintech use. In order of their presentation in our logit model table, we briefly look at what these variables are and what their coefficients indicate in terms of the relationship.

The only two features found significant from the demographic domain are "unemployed" as an occupation category and "traditional house" as a house category of the respondent. Both of these variables have negative coefficients, indicating unemployed respondents were significantly less likely to use fintech services; and the use of fintech services was less prevalent in rural or semi-urban areas compared to urban areas in our survey. As expected, we observe that coefficients for mobile use and lack of access to the internet were both significant, with positive and negative signs respectively. Our survey collected data on customers' use of a wide range of fintech services. Broadly, they were either accessed through mobile phones (e.g., mobile banking and payment services), and through a computer or internet banking application (e.g., remittance, deposit payment scheme installments). Most were users of mobile fintech services. While mobile banking

services in Bangladesh is accessible through mobile operators without direct access to internet on the user's end, availing more sophisticated fintech services, e.g., utility bill payment and load disbursement, required accessing the internet through a smartphone. Hence, a negative and statistically significant coefficient for lack of internet makes sense in the context of our data.

After Recursive Feature Elimination (RFE), 21 features survived from the customer concern class. Given they amount to 38.18% of total features included in the model, we conclude that reported customer concern on various issues constitute dominant part of the classification process. Of these 21 features, 14 were found to be significant at the 5% level. Overall, we observe that all statistically significant concern-related features, regardless of customers' reported levels on the Likert scale, were associated with negative coefficients. These concern features were related to "information secrecy", unknown issues", "limited government control" over emerging new fintech services and their operations with respect to customer welfare, "financial scandal" in fintech platforms, and "information security". In general, reported levels of "high" or "very high" concern on these issues were associated with larger negative coefficients. It is thus evident that higher levels of concern were prominent predictors of fintech usage, albeit negatively. This evidence supports findings in extant literature. As an example, Chowdhury and Hussain (2022) observed that perceived security of the system exerts strong influence on users for fintech adoption.

Customers' reported mental preparedness was found to be a significant predictor. Interestingly, among three levels of mental preparedness found to be significant, the largest negative coefficient was found for customers who reported to be "prepared" to use fintech services, followed by "not prepared at all". We explain this by noting that customers currently not using fintech services may have interpreted the question differently, thus overestimating mental preparedness for fintech use. Moreover, we observe that customers' perception of "geographic", "technological confidence" and "service intuitiveness" obstacles were significant predictors of fintech use. Customer feature of "very low" levels of reported geographic obstacles to fintech use was associated with a positive coefficient. Whereas, obstacle features related to "neutral" technological confidence and "high" service intuitiveness were associated with negatives. Indeed, Shareef et al. (2018) found evidence of perceived ability to use a service to have a significant influence on mobile banking adoption. We expect consumers' confidence in dealing with technology-driven services to have a major influence on perceived ability, thereby affecting adoption. Our results support this conjecture.

### 4.3.2. Practical Implication

The main implication of our findings for fintech service providers and regulators is to focus on customer perception in driving fintech adoption. More precisely, the design of intervention programs should primarily be informed by customer perception of obstacles, mental preparedness, etc., and customer concerns on security, privacy, and financial fraud issues while using fintech platforms. More than the demographic and economic profile of the target audience, these perception variables significantly determine adoption of fintech across Bangladesh. This design recommendation is true for both commercial market campaigns, as well as government programs, to raise awareness and drive fintech use. Our insight can help fintech service providers expand their user base more effectively in Bangladesh.

Mobile financial services have taken a firm foothold in Bangladesh during the last decade. The network of agent banking has expanded into increasingly remote locations (Hossain and Hossain 2015; Islam and Salma 2016; Siddik 2014). Contrary to theoretical expectations, we find geographic obstacles constitute a significant predictor of fintech use in our study. To what extent is this due to proximity to physical agent banking and other financial services networks or due to social driving factors of new technology use can be an interesting area of further investigation. Indeed, there is a growing body of literature on facilitating conditions, perceived variables, expectancies, social effects, personal factors

contributing to mobile banking adoption and use (Islam et al. 2017). Some authors have also investigated the moderating effect of demographic variables on this relationship.

In addition, the respondent feature of "high" level of satisfaction regarding fintech services exhibited lower negative but statistically significant coefficients compared to the respondent feature with no fintech use. Khan et al. (2021) found evidence of all dimensions of service quality connected to fintech satisfaction for customers in Bangladesh. In particular, the beta coefficient for responsiveness was strongest across dimensions, indicating service providers' responsiveness determine a large part of customer satisfaction for fintech in Bangladesh. When it comes to fintech service satisfaction, tangibility is a less significant factor (Khan et al. 2021).

Finally, service intuitiveness of the fintech product was found to be one of the significant obstacle features in our model. Azad (2016) used a neural network approach in investigating factors of adoption for mobile banking in Bangladesh. With robust 10-fold cross validated findings, ease of use of the mobile banking service was observed as the most important factor behind mobile banking adoption. Our evidence supports this finding. In this study, a wide range of respondent ages was incorporated. For older users who have recently shifted to fintech products, service intuitiveness is an important factor for fintech use. Particularly for new and more targeted fintech services, ease of use can determine whether customers adopt these offerings. Considering a combination of all the significant features and their relationship with fintech use, we conclude that respondents with access to mobile, lower levels of reported concerns with fintech use, average mental preparedness to use fintech services, and low levels of perceived geographic obstacle to fintech use were more likely to use fintech services. Conversely, respondents from semi-urban areas, high levels of reported concerns with fintech use, low levels of mental preparedness to use fintech services, high reported levels of technological confidence and service intuitiveness obstacle were less likely to use fintech services in our dataset.

## 5. Conclusions

To aid financial inclusion and financial resilience through innovative fintech solutions, understanding fintech adoption factors is important. The value of such insights is even more relevant for service providers and policy makers in an emerging economy like Bangladesh. Here, large parts of the population still remain unbanked or underbanked. This study was part of a wider series of research undertakings aimed to investigate fintech adoption, usage, readiness, and impact on sustainable economic development in Bangladesh. To that end, we conducted a nationwide representative survey and collected data on a wide range of demographic, economic, perceptual variables. We also collected data on technology use and banking activity. Our dataset also included respondents' fintech use and opinion related to concerns and obstacles faced.

Our binary logit model, estimated from selected features with fintech use as the dependent variable, yields important insights on contributing factors for the fintech user. We observe that fintech use is most prominently determined by customer security concerns and reported levels of obstacles faced with fintech use. Despite incorporating a wide range of demographic and economic variables, we find little evidence of influence of these factors from our dataset. We suggested that on a macro-level, fintech service providers, ecosystem enablers, and financial policy makers need to concentrate their efforts in addressing customer concerns and perceived obstacles. This can have, according to our findings, the biggest possible gain in facilitating wider adoption of customer-facing fintech services in Bangladesh, leading to greater access to financial services and better financial resilience of customers.

One of the key aspects of our study design was a nationwide representative sample covering a wide range of geographic, socio-economic, and demographic diversity. However, a central limitation of the study remains customer-reported data. Respondents were asked to rate their perception on mental preparedness, concerns, and perceived obstacles on Likert scales. The absence of an existing dataset make independent validation of these

ratings difficulty. Moreover, due to using recursive feature elimination (RFE), the authors had no control of over which domains of variables get relatively higher importance in the feature selection process based on existing theory. Instead, the authors took a bottom-up approach and tried to connect remaining features with implications thereof.

Future research can contribute in a number of ways. First, replication of our methodology for a comparable sample can be carried out in Bangladesh. In fact, the methodology can also be deployed in peer emerging markets experiencing a similar expansion in fintech adoption. They include Vietnam, Nigeria, Kenya, Pakistan, and Thailand, among others. Another area that future researchers can look into is the use of weighted methods where certain classes of variables can assume greater importance in the feature selection process. Future research may also delve deeper into effective ways to address customer concerns and perceived obstacles and assessing the impact of such intervention on fintech adoption intention.

**Author Contributions:** Conceptualization, K.M., M.M.A.J. and K.M.-U.-S.; Methodology, M.M.A.J. and K.M.-U.-S.; Software, K.M.; Formal analysis, K.M.; Investigation, M.M.A.J. and K.M.-U.-S.; Resources, M.M.A.J. and K.M.-U.-S.; Writing—original draft, K.M.; Writing—review & editing, M.M.A.J. and K.M.-U.-S.; Visualization, K.M.; Supervision, K.M.-U.-S.; Project administration, K.M. All authors have read and agreed to the published version of the manuscript.

**Funding:** This research was funded by the University of Dhaka under the Centennial Research Grant (CRG), and The APC was funded by Khaled Mahmud.

**Institutional Review Board Statement:** Not applicable.

**Informed Consent Statement:** During administration of questionnaire survey, all participants provided their explicit consent to the data collection process and approved use of the data for this research project.

**Data Availability Statement:** The dataset used in this paper was collected by the authors through a survey. The authors have not made the data publicly available for proprietary reasons.

**Conflicts of Interest:** The authors of the study do not find, to the best of their knowledge, any possible conflicts of interest.

## Appendix A

Bangladesh is divided into eight administrative divisions. We planned to collect data from two districts of each divisions. To select these, we looked into the distribution of Upazila/Metro Thana in poverty groups by district (Bangladesh Bureau of Statistics 2016). As we were looking for customer readiness regarding fintech, we wanted to consider all tiers of customers according to their financial situation to select our sample districts.

In the Table A1, we used a weight of 1 for very low poverty, 2 for low poverty, 3 for moderate, 4 for high, and 5 for very high rate of poverty. We calculated the weighted average score of each district. We then considered the lowest and highest scoring districts from each division. The bold-text rows in Table A1 shows selected districts from each division. In case of a tie, we selected randomly.

**Table A1.** District-wise weighted average score for sample construction.

| Division | District | Total | (1) | (2) | (3) | (4) | (5) | Weighted Average Score |
|---|---|---|---|---|---|---|---|---|
| Barisal | Barguna | 6 | 0 | 0 | 1 | 5 | 0 | 3.83 |
| | Barisal | 10 | 0 | 0 | 4 | 5 | 1 | 3.70 |
| | **Bhola** | **7** | **0** | **5** | **0** | **2** | **0** | **2.57** |
| | Jhalokati | 4 | 0 | 1 | 2 | 0 | 1 | 3.25 |
| | **Patuakhali** | **8** | **0** | **0** | **2** | **4** | **2** | **4.00** |
| | Pirojpur | 7 | 0 | 0 | 4 | 2 | 1 | 3.57 |
| Chittagong | Bandarban | 7 | 0 | 0 | 0 | 1 | 6 | 4.86 |
| | Brahmanbaria | 9 | 7 | 2 | 0 | 0 | 0 | 1.22 |
| | Chandpur | 8 | 0 | 0 | 3 | 3 | 2 | 3.88 |
| | Chittagong | 30 | 8 | 11 | 8 | 3 | 0 | 2.20 |
| | Comilla | 17 | 0 | 6 | 11 | 0 | 0 | 2.65 |
| | Cox's Bazar | 8 | 0 | 4 | 2 | 1 | 1 | 2.88 |
| | **Feni** | **6** | **5** | **1** | **0** | **0** | **0** | **1.17** |
| | **Khagrachhari** | **9** | **0** | **0** | **0** | **1** | **8** | **4.89** |
| | Lakshmipur | 5 | 0 | 0 | 2 | 1 | 2 | 4.00 |
| | Noakhali | 9 | 2 | 2 | 2 | 1 | 2 | 2.89 |
| | Rangamati | 10 | 0 | 0 | 1 | 1 | 8 | 4.70 |
| Dhaka | Dhaka | 55 | 45 | 8 | 1 | 1 | 0 | 1.24 |
| | Faridpur | 9 | 6 | 3 | 0 | 0 | 0 | 1.33 |
| | Gazipur | 13 | 7 | 6 | 0 | 0 | 0 | 1.46 |
| | Gopalganj | 5 | 0 | 0 | 3 | 2 | 0 | 3.40 |
| | **Kishoreganj** | **13** | **0** | **0** | **0** | **1** | **12** | **4.92** |
| | Madaripur | 4 | 4 | 0 | 0 | 0 | 0 | 1.00 |
| | Manikganj | 7 | 0 | 2 | 4 | 1 | 0 | 2.86 |
| | Munshiganj | 6 | 6 | 0 | 0 | 0 | 0 | 1.00 |
| | **Narayanganj** | **5** | **5** | **0** | **0** | **0** | **0** | **1.00** |
| | Narsingdi | 6 | 4 | 2 | 0 | 0 | 0 | 1.33 |
| | Rajbari | 5 | 0 | 0 | 2 | 3 | 0 | 3.60 |
| | Shariatpur | 6 | 0 | 4 | 2 | 0 | 0 | 2.33 |
| | Tangail | 12 | 0 | 3 | 7 | 2 | 0 | 2.92 |
| Khulna | Bagerhat | 9 | 0 | 2 | 6 | 1 | 0 | 2.89 |
| | Chuadanga | 4 | 0 | 0 | 3 | 1 | 0 | 3.25 |
| | Jessore | 8 | 0 | 0 | 4 | 4 | 0 | 3.50 |
| | Jhenaidah | 6 | 0 | 0 | 1 | 4 | 1 | 4.00 |
| | Khulna | 15 | 0 | 1 | 1 | 13 | 0 | 3.80 |
| | Kushtia | 6 | 1 | 3 | 2 | 0 | 0 | 2.17 |
| | **Magura** | **4** | **0** | **0** | **0** | **0** | **4** | **5.00** |
| | Meherpur | 3 | 0 | 0 | 2 | 0 | 1 | 3.67 |
| | **Narail** | **3** | **1** | **2** | **0** | **0** | **0** | **1.67** |
| | Satkhira | 7 | 1 | 6 | 0 | 0 | 0 | 1.86 |
| Mymensingh | **Jamalpur** | **7** | **0** | **0** | **0** | **0** | **7** | **5.00** |
| | **Mymensingh** | **13** | **0** | **1** | **5** | **6** | **1** | **3.54** |
| | Netrokona | 10 | 0 | 0 | 0 | 7 | 3 | 4.30 |
| | Sherpur | 5 | 0 | 0 | 0 | 3 | 2 | 4.40 |
| Rajshahi | Bogra | 12 | 1 | 2 | 5 | 4 | 0 | 3.00 |
| | Joypurhat | 5 | 0 | 2 | 3 | 0 | 0 | 2.60 |
| | Naogaon | 11 | 0 | 0 | 1 | 7 | 3 | 4.18 |
| | Natore | 7 | 0 | 3 | 2 | 2 | 0 | 2.86 |
| | **Chapai Nawabganj** | **5** | **0** | **0** | **0** | **1** | **4** | **4.80** |
| | Pabna | 9 | 0 | 0 | 6 | 1 | 2 | 3.56 |
| | **Rajshahi** | **15** | **3** | **10** | **1** | **1** | **0** | **2.00** |
| | Sirajganj | 9 | 0 | 0 | 3 | 6 | 0 | 3.67 |

**Table A1.** *Cont.*

| Division | District | Total | (1) | (2) | (3) | (4) | (5) | Weighted Average Score |
|---|---|---|---|---|---|---|---|---|
| | **Dinajpur** | **13** | **0** | **0** | **0** | **0** | **13** | **5.00** |
| | Gaibandha | 7 | 0 | 0 | 0 | 0 | 7 | 5.00 |
| | Kurigram | 9 | 0 | 0 | 0 | 0 | 9 | 5.00 |
| Rangpur | Lalmonirhat | 5 | 0 | 0 | 0 | 3 | 2 | 4.40 |
| | Nilphamari | 6 | 0 | 0 | 0 | 0 | 6 | 5.00 |
| | **Panchagarh** | **5** | **1** | **1** | **2** | **1** | **0** | **2.60** |
| | Rangpur | 8 | 0 | 0 | 0 | 4 | 4 | 4.50 |
| | Thakurgaon | 5 | 0 | 0 | 0 | 5 | 0 | 4.00 |
| | Habiganj | 9 | 1 | 6 | 2 | 0 | 0 | 2.11 |
| Sylhet | Moulvibazar | 7 | 3 | 2 | 2 | 0 | 0 | 1.86 |
| | **Sunamganj** | **11** | **0** | **6** | **3** | **1** | **1** | **2.73** |
| | **Sylhet** | **13** | **4** | **9** | **0** | **0** | **0** | **1.69** |

From each district, we selected Sadar (main) Upazilla[3] and the farthest Upazilla from Sadar Upazilla. In every Upazilla, we selected Sadar Union and the farthest Union from the Sadar Union. In each Union we selected the Sadar Ward and the farthest Ward from the Sadar Ward. In each Ward, we performed systematic sampling. From every ward we have collected 10 samples. We have divided the entire population of that Ward by 10. Then we have randomly started form one house to the every $n$th ($n$ = population of the ward/10) house. From every union we collected 20 samples. From every Upazilla, we collected 40 samples. As a result, from every district we collected 80 samples. From 16 districts, we have collected 1282 samples in total.

**Appendix B**

The following Table A2 lists all variables used in our model. In case of categorical variables, a list of levels is also given.

**Table A2.** List of variables used in the model and their levels.

| Variable | Levels |
|---|---|
| Gender | Male, Female |
| Age | - |
| Education | Primary, Secondary, None, Higher secondary, Graduate, Post-graduate, Madrasa_(kawmi) |
| Marriage | Married, Single |
| Occupation | Business, Day Laborer, Homemaker, Non-government Job, Retired, Student, Unemployed, Driver (Rickshaw/Van/Engine Vehicle), Farmer/Fisherman/Boatman, Government Job, Government Allowance, Non-resident. Others |
| Household | - |
| Expenses | - |
| ExpRent | - |
| ExpFood | - |
| ExpUtilities | - |
| ExpEducation | - |
| ExpHealthcare | - |
| ExpEntertainment | - |
| ExpClothing | - |
| ExpHouseHelp | - |

**Table A2.** *Cont.*

| Variable | Levels |
|---|---|
| ExpMisc | - |
| Income | - |
| AnnualSaving | - |
| House | Traditional House, Cemented House |
| BankAccount | No, Yes |
| BankVisit | - |
| BankAwareness | Very low knowledge (only deposited and withdrawal), Some knowledge (deposited scheme and loan scheme), No knowledge at all, Above average knowledge (LC, stock market, financial report, ratios etc.), Expert (certified financial analyst) |
| Computer | No, Yes |
| Mobile | No, Yes |
| SmartphoneSkill | Not skilled at all, Very low skills, Some skills, Skilled, Very skilled |
| Internet | No, Yes |
| Data_usage | - |
| Concern_Information_Secrecy | I don't Know, Very Low, Low, More or less, High, Very High |
| Concern_Unknown_Issues | I don't Know, Very Low, Low, More or less, High, Very High |
| Concern_Limited_GovControl | I don't Know, Very Low, Low, More or less, High, Very High |
| Concern_Financial_Scandal | I don't Know, Very Low, Low, More or less, High, Very High |
| Concern_Cashless_Community | I don't Know, Very Low, Low, More or less, High, Very High |
| Concern_Information_Security | I don't Know, Very Low, Low, More or less, High, Very High |
| MentalPreparedness | Low prepared, Not prepared at all, Average preparedness, Prepared, Adequately prepared |
| Fintech_satisfaction | I don't use fintech, Satisfied, Neutral, Dissatisfied, Highly dissatisfied, Highly satisfied |
| Max_fee_per_1000 | - |
| Obstacle_economic_condition | Very low, Low, Neutral, High, Very high |
| Obstacle_geographic_location | Very low, Low, Neutral, High, Very high |
| Obstacle_confidence_in_technolog | ery low, Low, Neutral, High, Very high |
| Obstacle_service_intuitiveness | Very low, Low, Neutral, High, Very high |
| Fintech_service_affordability | Very low, Low, Neutral, High, Very high |
| Fintech_costliness | I don't know, Not affordable at all, Not affordable, Neutral, Affordable, Highly affordable |

**Appendix C**

The following table lists all features, associated coefficients, *p*-values, and levels of significance. Coefficients significant at the 5% level ($p \leq 0.05$) are marked with "***". The logit model was estimated with Maximum Likelihood Estimate with the binary dependent variable "FintechUser". The model covariances are no-robust. McFadden's pseudo-R Squared is 0.6765.

**Table A3.** Logistic regression results and feature significance levels.

| Feature | Coef. | Std. Err. | z-Value | p-Value | [95% Conf. | Interval] | Sig. |
|---|---|---|---|---|---|---|---|
| Gender_Male | 0.242 | 0.612 | 0.395 | 0.693 | −0.958 | 1.442 | |
| Education_Madrasa_(kawmi) | 1.535 | 0.93 | 1.65 | 0.099 | −0.288 | 3.358 | |
| Marriage_Married | −0.73 | 0.373 | −1.957 | 0.05 | −1.461 | 0.001 | |
| Occupation_Government Allowance | −21.912 | 31200 | −0.001 | 0.999 | −61,100 | 61,100 | |
| Occupation_Homemaker | −1.065 | 0.693 | −1.536 | 0.124 | −2.424 | 0.294 | |
| Occupation_Non-government Job | 0.669 | 0.388 | 1.723 | 0.085 | −0.092 | 1.431 | |
| Occupation_Others | −1.076 | 0.863 | −1.246 | 0.213 | −2.768 | 0.617 | |
| Occupation_Retired | −0.877 | 0.865 | −1.015 | 0.31 | −2.572 | 0.817 | |
| Occupation_Student | 0.527 | 0.571 | 0.924 | 0.356 | −0.591 | 1.646 | |
| Occupation_Unemployed | −1.208 | 0.581 | −2.081 | 0.037 | −2.346 | −0.07 | *** |
| House_Traditional House | −0.69 | 0.262 | −2.63 | 0.009 | −1.203 | −0.176 | *** |
| BankAccount_No | −0.372 | 0.266 | −1.399 | 0.162 | −0.892 | 0.149 | |
| BankAwareness_Above average knowledge (LC, stock market, financial report, ratios etc. | 2.103 | 1.601 | 1.313 | 0.189 | −1.035 | 5.242 | |
| BankAwareness_Expert (certified finanical analyst) | −42.155 | 832,000,000 | 0 | 1 | −1,630,000,000 | 1,630,000,000 | |
| BankAwareness_Some knowlede (deposite scheme and loan scheme) | −0.628 | 0.328 | −1.914 | 0.056 | −1.272 | 0.015 | |
| Mobile_No | −15.358 | 36,300 | 0 | 1 | −71,200 | 71,200 | |
| Mobile_Yes | 9.052 | 1.271 | 7.122 | 0 | 6.561 | 11.543 | *** |
| Internet_No | −0.944 | 0.262 | −3.599 | 0 | −1.458 | −0.43 | *** |
| Concern_Information_Secrecy_High | −1.621 | 0.474 | −3.422 | 0.001 | −2.549 | −0.692 | *** |
| Concern_Information_Secrecy_Low | −1.23 | 0.511 | −2.408 | 0.016 | −2.23 | −0.229 | *** |
| Concern_Information_Secrecy_More or less | −0.748 | 0.467 | −1.602 | 0.109 | −1.663 | 0.167 | |
| Concern_Unknown_Issues_I don't Know | −0.834 | 0.853 | −0.978 | 0.328 | −2.506 | 0.837 | |
| Concern_Unknown_Issues_Very High | −0.749 | 0.562 | −1.333 | 0.183 | −1.851 | 0.353 | |
| Concern_Unknown_Issues_Very Low | −1.143 | 0.809 | −1.413 | 0.158 | −2.728 | 0.443 | |
| Concern_Limited_GovControl_High | −1.961 | 0.809 | −2.424 | 0.015 | −3.546 | −0.375 | *** |
| Concern_Limited_GovControl_I don't Know | −1.358 | 1.084 | −1.253 | 0.21 | −3.483 | 0.766 | |
| Concern_Limited_GovControl_Low | −1.61 | 0.824 | −1.953 | 0.051 | −3.225 | 0.006 | |
| Concern_Limited_GovControl_More or less | −1.811 | 0.813 | −2.227 | 0.026 | −3.404 | −0.217 | *** |
| Concern_Limited_GovControl_Very High | −2.365 | 0.894 | −2.645 | 0.008 | −4.116 | −0.613 | *** |
| Concern_Financial_Scandal_I don't Know | −2.853 | 0.636 | −4.489 | 0 | −4.099 | −1.607 | *** |
| Concern_Financial_Scandal_More or less | −1.338 | 0.343 | −3.897 | 0 | −2.011 | −0.665 | *** |
| Concern_Financial_Scandal_Very Low | −2.718 | 1.346 | −2.019 | 0.044 | −5.357 | −0.079 | *** |
| Concern_Cashless_Community_High | −0.545 | 0.312 | −1.747 | 0.081 | −1.157 | 0.066 | |
| Concern_Cashless_Community_Very High | −1.064 | 0.492 | −2.161 | 0.031 | −2.029 | −0.099 | *** |
| Concern_Information_Security_High | −2.326 | 0.831 | −2.798 | 0.005 | −3.955 | −0.696 | *** |
| Concern_Information_Security_I don't Know | −2.177 | 1.034 | −2.106 | 0.035 | −4.203 | −0.151 | *** |
| Concern_Information_Security_Low | −2.437 | 0.837 | −2.91 | 0.004 | −4.078 | −0.795 | *** |
| Concern_Information_Security_More or less | −2.362 | 0.844 | −2.797 | 0.005 | −4.017 | −0.707 | *** |
| Concern_Information_Security_Very High | −1.832 | 0.89 | −2.059 | 0.04 | −3.576 | −0.088 | *** |

**Table A3.** *Cont.*

| Feature | Coef. | Std. Err. | z-Value | p-Value | [95% Conf. | Interval] | Sig. |
|---|---|---|---|---|---|---|---|
| MentalPreparedness_Average preparedness | −0.931 | 0.291 | −3.199 | 0.001 | −1.501 | −0.36 | *** |
| MentalPreparedness_Not prepared at all | −1.181 | 0.595 | −1.986 | 0.047 | −2.347 | −0.015 | *** |
| MentalPreparedness_Prepared | −1.653 | 0.377 | −4.387 | 0 | −2.392 | −0.915 | *** |
| Fintech_satisfaction_Highly satisfied | −1.309 | 0.763 | −1.715 | 0.086 | −2.805 | 0.187 | |
| Fintech_satisfaction_I don't use fintech | −2.487 | 0.487 | −5.107 | 0 | −3.441 | −1.533 | *** |
| Fintech_satisfaction_Neutral | 0.47 | 0.291 | 1.616 | 0.106 | −0.1 | 1.039 | |
| Obstacle_geographic_location_High | −1.425 | 0.546 | −2.609 | 0.009 | −2.496 | −0.355 | *** |
| Obstacle_geographic_location_Very high | −0.799 | 1.182 | −0.676 | 0.499 | −3.117 | 1.518 | |
| Obstacle_geographic_location_Very low | 1.045 | 0.421 | 2.484 | 0.013 | 0.22 | 1.87 | *** |
| Obstacle_confidence_in_technolog_Neutral | −0.704 | 0.254 | −2.771 | 0.006 | −1.202 | −0.206 | *** |
| Obstacle_service_intuitiveness_High | −1.322 | 0.542 | −2.44 | 0.015 | −2.384 | −0.26 | *** |
| Obstacle_service_intuitiveness_Low | −0.893 | 0.486 | −1.838 | 0.066 | −1.846 | 0.059 | |
| Obstacle_service_intuitiveness_Neutral | −0.888 | 0.485 | −1.832 | 0.067 | −1.838 | 0.062 | |
| Fintech_service_affordability_Highly affordable | 1.383 | 1.229 | 1.125 | 0.26 | −1.026 | 3.792 | |
| Fintech_service_affordability_I don't know | −0.816 | 0.748 | −1.091 | 0.275 | −2.282 | 0.65 | |
| Fintech_service_affordability_Not affordable | −0.6 | 0.311 | −1.93 | 0.054 | −1.21 | 0.009 | |

*** denotes significance at the 5% level.

## Notes

[1] South Asian Association for Regional Cooperation.
[2] Association of Southeast Asian Nations.
[3] Upazila is an administrative smaller than districts.

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
