# Peer review of "Adoption Factors of FinTech: Evidence from an Emerging Economy Country-Wide Representative Sample"

_ijfs, doi:10.3390/ijfs11010009_

Round 1

Reviewer 1 Report

- The main issue is the definition or scope of FinTech in this paper, therefore authors are suggested to provide a proper definition of FinTech (specifically in the context of Bangladesh). FinTech is a very broad term.

- Further, authors are suggested to provide a questionnaire in the appendix, if possible.

- The objective and model of the study are not clear, therefore, it is needed to provide a clear model (dependent and independent and other variables in one diagram).

- Professional proofreading of the draft is required.

Reviewer 2 Report

The paper presents a topic that is very interesting, readable and of interest today. The digitalisation of finance and the emergence of new financial players have been in the spotlight in recent years. I therefore consider the paper to be a niche that is likely to be of great interest.
I have no suggestions for changes in the formal presentation of the paper, as it is on the whole a well-edited, clear and, in terms of the way the chapters are structured, a high-quality paper. The number of literature sources used is adequate and relevant literature has been developed by the authors. From a content point of view, what I would suggest would be a little more in-depth statistical analysis (as the data set allows), but this is only a suggestion. The methodology used is still comprehensible and can be followed by readers who are familiar with the subject or who are only interested in it for the time being.

Author Response

We sincerely thank the reviewer for their insightful comments and inspiration. Given the main focus of the paper is modeling fintech adoption, the statistical analysis in section 4.1.1 through 4.1.7 is an incidental addition. This is aimed at furthering the reader's understanding of fintech use across different groups in our sample. We think further expansion here is not strictly necessary. 

Reviewer 3 Report

Review report

 Journal: IJFS

 Manuscript ID: ijfs-2074977

 Title: “Adoption Factors of FinTech: Evidence from an Emerging Economy Country-wide Representative Sample”

Overall

• Authors should strive to explain why this research is important and what motivated them to conduct this research in more detail and in a more convincing manner.

• Further, the paper’s theoretical contribution is vague and not sufficiently highlighted.

• I suggest the authors to add a special section in contributions to the study: 1. Theoretical contribution 2. Practical implication.

• The authors should present the limitations precisely in the relevant section.

• There should not be any abbreviation in abstract.

• Table No. (3) lacks arrangement, consistency with other table, please pay attention.

• Please pay attention to filling in the following sections at the end of the research (Author Contributions, Acknowledgments, etc.)

Introduction

·         In my view, the results should not be mentioned in the introduction “We find that respondents with mobile access and lower levels of reported concerns with security, as well as low levels of reported geographic obstacle are more likely to use fintech services. Whereas, respondents with high levels of concerns for security and finan- cial scam issues on fintech services, low levels of confidence using new technological so- lutions, and high reported levels of obstacle with service intuitiveness are less likely to use fintech services.”

·         I think it is appropriate to briefly explain or explain FinTech applications (such as digital banking services, electronic wallets, digital payments and transfers, digital insurance, crowdfunding) that the study aimed to focus on in the introduction. The authors should explain to the readers their procedural definition of fintech adoption. Fintech adoption in general. Or digital banking, or the adoption of financial technology in the field of insurance, for example!

·         Again, the authors mention fintech adoption in the text. Please define the procedural concept of financial technology or the adoption of financial technology. Financial technology is not limited only to banks and digital banking services!

Literature review

·         The authors’ claim that the Fintech is important due to its significant contribution to the Financial inclusion is insufficient. Please elaborate further.

·         I think it is more appropriate to use the term "Literature review" instead of "review of Literature"

Methodology

·         The authors should provide how the questionnaire was distributed. Furthermore, the authors should explain how the questionnaire was compiled after it was filled out.

·         The limitation of the sampling method should be mentioned in the limitations section.

·         Please explain more about the details of the survey.

·         The response rate seems high regarding the exiting literature dealing with survey methods. Were there any incentives to participate in this survey? Please explain.

·         “experts’ opinions were also obtained”: Please elaborate.

·         I think that the program that was used in the statistical analysis needs more clarification.

Discussion

·         The authors should expand in elaborate on the findings of their research and compare it with the results of other research.

Limitations

·         All the comments, concerns, and suggestions should be presented in this section.

Conclusion: I was hoping to find recommendations for regulators, fintech companies, and financial institutions in Bangladesh at the end of this research.

                                  With sincere wishes of success to the authors!

Review report

 Journal: IJFS

 Manuscript ID: ijfs-2074977

 Title: “Adoption Factors of FinTech: Evidence from an Emerging Economy Country-wide Representative Sample”

Overall

• Authors should strive to explain why this research is important and what motivated them to conduct this research in more detail and in a more convincing manner.

• Further, the paper’s theoretical contribution is vague and not sufficiently highlighted.

• I suggest the authors to add a special section in contributions to the study: 1. Theoretical contribution 2. Practical implication.

• The authors should present the limitations precisely in the relevant section.

• There should not be any abbreviation in abstract.

• Table No. (3) lacks arrangement, consistency with other table, please pay attention.

• Please pay attention to filling in the following sections at the end of the research (Author Contributions, Acknowledgments, etc.)

Introduction

·         In my view, the results should not be mentioned in the introduction “We find that respondents with mobile access and lower levels of reported concerns with security, as well as low levels of reported geographic obstacle are more likely to use fintech services. Whereas, respondents with high levels of concerns for security and finan- cial scam issues on fintech services, low levels of confidence using new technological so- lutions, and high reported levels of obstacle with service intuitiveness are less likely to use fintech services.”

·         I think it is appropriate to briefly explain or explain FinTech applications (such as digital banking services, electronic wallets, digital payments and transfers, digital insurance, crowdfunding) that the study aimed to focus on in the introduction. The authors should explain to the readers their procedural definition of fintech adoption. Fintech adoption in general. Or digital banking, or the adoption of financial technology in the field of insurance, for example!

·         Again, the authors mention fintech adoption in the text. Please define the procedural concept of financial technology or the adoption of financial technology. Financial technology is not limited only to banks and digital banking services!

Literature review

·         The authors’ claim that the Fintech is important due to its significant contribution to the Financial inclusion is insufficient. Please elaborate further.

·         I think it is more appropriate to use the term "Literature review" instead of "review of Literature"

Methodology

·         The authors should provide how the questionnaire was distributed. Furthermore, the authors should explain how the questionnaire was compiled after it was filled out.

·         The limitation of the sampling method should be mentioned in the limitations section.

·         Please explain more about the details of the survey.

·         The response rate seems high regarding the exiting literature dealing with survey methods. Were there any incentives to participate in this survey? Please explain.

·         “experts’ opinions were also obtained”: Please elaborate.

·         I think that the program that was used in the statistical analysis needs more clarification.

Discussion

·         The authors should expand in elaborate on the findings of their research and compare it with the results of other research.

Limitations

·         All the comments, concerns, and suggestions should be presented in this section.

Conclusion: I was hoping to find recommendations for regulators, fintech companies, and financial institutions in Bangladesh at the end of this research.

                                  With sincere wishes of success to the authors!

Round 2

Reviewer 1 Report

The suggested comments are incorporated.